# Assessment of the Proton Pump Inhibitor, Esomeprazole Magnesium Hydrate and Trihydrate, on Pathophysiological Markers of Preeclampsia in Preclinical Human Models of Disease

**DOI:** 10.3390/ijms23179533

**Published:** 2022-08-23

**Authors:** Natasha de Alwis, Bianca R. Fato, Sally Beard, Natalie K. Binder, Tu’uhevaha J. Kaitu’u-Lino, Kenji Onda, Natalie J. Hannan

**Affiliations:** 1Therapeutics Discovery and Vascular Function Group, Department of Obstetrics and Gynaecology, Mercy Hospital for Women, University of Melbourne, Heidelberg, VIC 3084, Australia; 2Diagnostics Discovery and Reverse Translation in Pregnancy Group, Department of Obstetrics and Gynaecology, Mercy Hospital for Women, University of Melbourne, Heidelberg, VIC 3084, Australia; 3Department of Clinical Pharmacology, Tokyo University of Pharmacy and Life Sciences, Hachioji, Tokyo 192-0392, Japan

**Keywords:** preeclampsia, placenta, esomeprazole, trophoblast, sFLT-1

## Abstract

Previously, we demonstrated that the proton pump inhibitor, esomeprazole magnesium hydrate (MH), could have potential as a repurposed treatment against preeclampsia, a serious obstetric condition. In this study we investigate the difference in the preclinical effectiveness between 100 µM of esomeprazole MH and its hydration isomer, esomeprazole magnesium trihydrate (MTH). Here, we found that both treatments reduced secretion of sFLT-1 (anti-angiogenic factor) from primary cytotrophoblast, but only esomeprazole MH reduced sFLT-1 secretion from primary human umbilical vein endothelial cells (assessed via ELISA). Both drugs could mitigate expression of the endothelial dysfunction markers, vascular cell adhesion molecule-1 and endothelin-1 (via qPCR). Neither esomeprazole MH nor MTH quenched cytotrophoblast reactive oxygen species production in response to sodium azide (ROS assay). Finally, using wire myography, we demonstrated that both compounds were able to induce vasodilation of human omental arteries at 100 µM. Esomeprazole is safe to use in pregnancy and a candidate treatment for preeclampsia. Using primary human tissues and cells, we validated that esomeprazole is effective in enhancing vascular relaxation, and can reduce key factors associated with preeclampsia, including sFLT-1 and endothelial dysfunction. However, esomeprazole MH was more efficacious than esomeprazole MTH in our in vitro studies.

## 1. Introduction

Preeclampsia is a complex, serious condition of pregnancy, affecting 3–8% of pregnancies worldwide, and is responsible for significant maternal and perinatal morbidity and mortality [1]. Preeclampsia can stem from a dysfunctional, ischemic placenta, which releases an excess of pro-inflammatory and anti-angiogenic factors, in particular sFLT-1, into the maternal circulation causing widespread maternal endothelial dysfunction and systemic vasoconstriction, culminating in hypertension and end-organ injury [1,2].

Unfortunately, there is still no cure for preeclampsia and limited treatment options are available. Delivery remains the mainstay treatment, as removal of the placenta withdraws the source of the circulating factors driving maternal injury, and maternal symptoms cease [1,3]. However, if disease ensues early, delivering the foetus preterm can come with its own serious complications for the neonate. Hence, there is an urgent unmet need to discover and test treatments that quench disease to prolong gestation, improving both maternal and foetal outcomes.

Proton pump inhibitors (PPIs) are a class of drugs predominantly prescribed to prevent and treat stomach and intestinal ulcers and gastroesophageal reflux disease. Large epidemiological studies have clearly shown that PPIs are safe for use in pregnancy [4,5].

Fundamental work carried out by our team previously demonstrated that PPIs can mediate multiple aspects of the pathophysiology of preeclampsia. In in vitro studies, the PPIs esomeprazole, lansoprazole, rabeprazole, omeprazole, and pantoprazole significantly reduced secretion of anti-angiogenic factors, including sFLT-1 (which is markedly upregulated in preeclampsia) [6,7] from primary human placental tissue and cells, including endothelium, from both normal and preeclamptic pregnancies. Esomeprazole, rabeprazole, and lansoprazole reduced markers of endothelial dysfunction and upregulated antioxidant cytoprotective molecules. Esomeprazole treatment reduced cytokine release from placenta and endothelial cells. Using ex vivo studies, we demonstrated that esomeprazole directly induced vasodilation of omental arteries and reduced blood pressure in a mouse model of preeclampsia [8]. Subsequently we explored the efficacy of combining esomeprazole with other potential candidate therapies and demonstrated that combining esomeprazole treatment with the candidate preeclampsia therapeutics, metformin or sulfasalazine, additively reduced sFLT-1 secretion and markers of endothelial dysfunction [9,10]. Collectively, our pre-clinical studies suggest that the PPIs, particularly esomeprazole, could be effective in treating preeclampsia.

These studies prompted us and others to test the efficacy of esomeprazole to treat preeclampsia in randomised controlled trials [10]. However, esomeprazole administered in the form of daily oral 40 mg tablets did not prolong gestation, nor decrease circulating sFLT-1 levels [11,12]. Circulating drug levels may not have been sufficient to exert clinical effects. A study by Saleh et al. investigated sFLT-1 levels among women taking PPIs, where women more likely took the PPI throughout the pregnancy, revealing reduced sFLT-1 and other pathogenic factors [13]. Further clinical phase II and III trials are currently examining esomeprazole in Australia (ACTRN12618001755224) and Egypt (NCT03717740; NCT03213639).

Esomeprazole magnesium hydrate (MH) was the compound used to assess the effect of esomeprazole in preclinical trials [8]. However, esomeprazole is commonly used in the clinic in the form of esomeprazole magnesium *trihydrate* (MTH); differences in the pharmacological potency between the hydrate isomers has not been previously investigated. Hence, in this study we aimed to investigate whether these two formulations may differ in action in vitro and ex vivo, particularly as hydration isomers can have different stability, solubility and bioavailability [14]. Importantly, we examined side-by-side the ability of these two esomeprazole hydrate isomers to: (1) reduce secretion of anti-angiogenic factor sFLT-1, (2) mediate endothelial dysfunction and oxidative stress, and (3) induce vasodilation; key aspects of the pathogenesis of preeclampsia.

## 2. Results

### 2.1. Esomeprazole Magnesium Hydrate and Esomeprazole Magnesium Trihydrate Decrease sFLT-1 Expression and Secretion

In preeclampsia, excess secretion of sFLT-1 is thought to be central to the endothelial dysfunction that drives vasoconstriction, leading to the clinical features of hypertension, and subsequent maternal organ injury. Similar to previous findings [8], here we found that primary cytotrophoblast treatment with esomeprazole magnesium hydrate (MH) demonstrated reduced secretion of sFLT-1 (Figure 1A, *p* = 0.0012), as well as reduced mRNA expression of the two variants *sFLT1-e15a* (Figure 1B, *p* < 0.0001) and *sFLT1-i13* (Figure 1C, *p* < 0.0001). Likewise, here we show that treatment of primary cytotrophoblast with the alternate compound, esomeprazole magnesium trihydrate (MTH), demonstrated a consistent decrease in mRNA expression of *sFLT1-e15a* (Figure 1B, *p* < 0.0001) and *sFLT1-i13* (Figure 1C, *p* < 0.0001). The secretion of sFLT-1 by cytotrophoblast was also significantly reduced following treatment with esomeprazole MTH (Figure 1A; *p* = 0.0361).

The endothelium is another major source of sFLT-1. In our previous study, esomeprazole MH reduced sFLT-1 secretion but not expression of the two sFLT-1 variant isoforms. In the current study, when administered to human umbilical vein endothelial cells (HUVECs), we again found esomeprazole MH reduced the secretion of sFLT-1 by HUVECs (Figure 1D, *p* = 0.0411), but esomeprazole MTH had no significant effect. Neither esomeprazole MH nor esomeprazole MTH altered the mRNA expression of either *sFLT1-e15a* (Figure 1E) or *sFLT1-i13* (Figure 1F) variants.

### 2.2. Both Esomeprazole Hydrate Isomers Rescue Endothelial Dysfunction In Vitro

Given that endothelial dysfunction is central to the pathogenesis of preeclampsia [3,15], another therapeutic strategy may be to identify drugs that act at the level of the endothelium to reduce dysfunction. Previously, we demonstrated that esomeprazole MH potently mediated endothelial dysfunction [8]. Here, we directly compared both esomeprazole MH and MTH and examined whether esomeprazole MTH can also mediate endothelial dysfunction by examining markers of dysfunction (vascular cell adhesion molecule-1 (*VCAM-1)* and endothelin-1 (*ET-1*)). Consistent with our previous findings, TNF-α, a pro-inflammatory cytokine that circulates in excess in preeclampsia [16] and a likely contributor to endothelial dysfunction [17], successfully induced endothelial dysfunction (5), indicated by the significant increase in expression of *ET-1* (Figure 2A; *p* < 0.0001) and *VCAM1* (Figure 2B; *p* < 0.0001).

Similar to our previous study [8], here we found addition of esomeprazole MH to HUVECs reduced TNF-α-induced *ET-1* mRNA expression (Figure 2A, *p* < 0.0001), and we observed the same for esomeprazole MTH, which also demonstrated significantly reduced mRNA expression of *ET-1* (Figure 2A, *p* < 0.0001). Likewise, treatment of HUVECs with esomeprazole MH (Figure 2B, *p* < 0.0001) and esomeprazole MTH (Figure 2B, *p* < 0.0001) potently mediated TNF-α-induced upregulation of *VCAM1*.

### 2.3. Esomeprazole Magnesium Hydration Isomers Do Not Mitigate Sodium-Azide-Induced Oxidative Stress in Human Primary Cytotrophoblasts

Excessive oxidative stress and inflammation is likely to exacerbate the placental and endothelial dysfunction that occurs in preeclampsia [18]. Therefore, reducing oxidative stress and inflammation may help decrease disease severity. Here, we assessed whether pre-incubation of primary cytotrophoblast in either esomeprazole MTH or esomeprazole MH could mitigate production of reactive oxygen species (ROS) induced by sodium azide (Na Azide). Cytotrophoblasts pre-incubated with esomeprazole MTH or esomeprazole MH demonstrated no significant change in ROS production in the presence of Na Azide, compared with cells pre-incubated in the vehicle control (Figure 3). These data demonstrate that neither esomeprazole compound can reduce the induction of ROS by Na Azide in primary cytotrophoblasts.

### 2.4. Esomeprazole Magnesium Hydrate and Esomeprazole Magnesium Trihydrate Induced Vasodilation of Whole Pregnant Human Arteries Ex Vivo

Systemic vasoconstriction and hypertension are hallmark features of preeclampsia. We next examined whether esomeprazole MTH had vasoactive properties. Maternal omental arteries were isolated from omental fat tissue obtained at caesarean section from healthy individuals at term. Wire myograph ex vivo experiments were performed. Both esomeprazole MTH (*p* = 0.0012) and esomeprazole MH (*p* < 0.0001) treatment induced significant relaxation of pre-constricted arteries compared with control (vehicle) (Figure 4A), when administered at the highest concentration (100 μmol/L). Further, the maximum relaxation achieved by esomeprazole treatment compared with the control was determined. The maximum relaxation (Rmax; Figure 4B) was not reduced by esomeprazole compared with the control; noting the controls and treated arteries reached maximum relaxation at different concentrations. We also investigated the area under the curve that defines the total relaxation over the experiment. We observed no significant difference in the area under the curve with either esomeprazole MTH or esomeprazole MH treatment (data not shown).

## 3. Discussion

Our group previously explored the effectiveness of several proton pump inhibitors to mitigate key pathogenic actions in pre-clinical models of preeclampsia [8], identifying esomeprazole to be the most efficacious. In the current study, we set out to investigate whether differences in efficacies exist between two esomeprazole hydrate compounds, esomeprazole MH (as used in our original study [8]) and esomeprazole MTH. This may have relevance in translation and clinical trials, as the two hydration isomers differ in solubility, and thus bioavailability in vivo. We did not expect an overt difference in efficacy, as the pharmacology of the active esomeprazole molecule should be the same in each compound. Here, we explored their ability to reduce several pathophysiological aspects of preeclampsia, scrutinising the effectiveness between the two. Overall, of the two esomeprazole compounds, esomeprazole MH appears more efficacious in mitigating pathogenic actions in our pre-clinical models of preeclampsia.

In the current study, we examined both esomeprazole compounds’ ability to reduce the expression and secretion of the pathogenic anti-angiogenic factor, sFLT-1, since it is dramatically elevated in the maternal circulation in preeclampsia [2,19,20,21]. Thus, sFLT-1 is a target molecule in preeclampsia therapeutic development.

In our previous study where we performed the first examination of proton pump inhibitors in models of preeclampsia, we found esomeprazole (MH) dose-dependently reduced secretion of sFLT-1 and the expression of its mRNA variants in primary placental tissue, cytotrophoblast cells [8]. Consistently in this study, esomeprazole MH reduced secretion of sFLT-1 by primary cytotrophoblast and HUVECs, whereas at a matched concentration, esomeprazole MTH only reduced sFLT-1 secretion in cytotrophoblast; however, there was a clear trend for reduction in the HUVECs. Further, both esomeprazole MH and MTH reduced mRNA expression of both sFLT-1 variants, *sFLT1-e15a* and *sFLT1-i13,* in primary cytotrophoblast. As we observed previously, despite a reduction in sFLT-1 protein secretion in primary HUVECs with treatment of esomeprazole MH only, neither of the esomeprazole compounds were able to reduce expression of either of the sFLT-1 variant mRNAs in HUVECs. Collectively, these data taken together suggest that esomeprazole MH is more potent in reducing sFLT-1 in our models. However, we note that these studies are performed in a relatively small sample size, and because esomeprazole MTH does trend the same way, expanding the sample size may produce a significant effect.

In addition, we explored key markers of endothelial dysfunction, *ET-1* and *VCAM1*, both known to be elevated in preeclamptic pregnancies [22,23,24,25,26,27]. ET-1 is a potent vasoconstrictor released from the endothelium into the circulation, and VCAM1 acts to trap and recruit inflammatory immune cells on the surface endothelium. Transcript expression of *ET-1* and *VCAM-1* was reduced by both esomeprazole MH and MTH. This also validates our previous findings, where esomeprazole MH reduced endothelial dysfunction in multiple assays [8]. Thus, both esomeprazole compounds quench markers of endothelial dysfunction in vitro.

We next interrogated whether esomeprazole could prevent or reduce the production of reactive oxygen species (ROS) by cytotrophoblast cells that contribute to oxidative stress. Placental oxidative stress and inflammation are key characteristics in the pathophysiology of preeclampsia [18] and hence are targets for therapeutic assessment. Pre-treatment with esomeprazole MH and esomeprazole MTH had no effect on sodium-azide-induced ROS. This indicates that both esomeprazole compounds could not mediate oxidative stress in the placental cells in our in vitro model of preeclampsia.

Finally, we set out to determine the ability of both esomeprazole MH and MTH to induce vascular dilation. Previously, we showed that esomeprazole MH induced potent endothelium-dependent vasodilation in human omental arteries collected from normotensive, term pregnancies or pregnancies complicated by early-onset preeclampsia, using pressure myography [8]. In the current study, we used a different vascular reactivity technique, *wire* myography, to assess the vascular effects of both forms of esomeprazole. Both formulations were successful in inducing vasodilation of vessels obtained from normotensive pregnancies ex vivo, validating our previous findings. Hence, this study demonstrates that both esomeprazole MH and MTH can drive vasodilatory effects in whole vessels using a distinct vascular reactivity assay, building on our previous data using esomeprazole MH.

Here, we performed functional assays utilizing several types of gestational tissue/cells, including whole human resistance arteries, primary endothelial cells, and primary cytotrophoblast cells. Our primary aim was to compare the efficacy between the two esomeprazole compounds in our human models, exploring therapeutic development targeted at the pathophysiology of preeclampsia for future translation. The two formulations of esomeprazole used in this study are hydration isomers, differing only in two hydrate groups. This would be expected to influence the solubility of these drugs, but this effect should disappear when the drugs are in solution. Further, all experiments controlled for the solvents that the drugs were reconstituted in, eliminating any confounding effects. However, our in vitro studies may partly be limited by our relatively small sample sizes. Further studies focused on identification of the molecular mechanisms are still important. Hydration isomers may also differ in stability and bioavailability [14]; this could be of particular importance when considering clinical trials. Additionally, future studies may elucidate whether the effect of these two compounds may differ from other formulations, such as esomeprazole sodium, which may have different clinical benefits since it is administered to patients intravenously versus the oral administration of esomeprazole magnesium, bypassing the gut.

Overall, both esomeprazole compounds demonstrate key actions towards mitigating the pathophysiology of preeclampsia in primary human preclinical assays. However, esomeprazole MH had additional benefits in reducing endothelial sFLT-1 secretion. Further pharmacokinetic and metabolic studies are required to determine if these data provide important information for consideration in clinical trials.

## 4. Materials and Methods

### 4.1. Tissue Collection

Ethical approval was obtained from the Mercy Health Human Research Ethics Committee. Women presenting to the Mercy Hospital for Women (Heidelberg, Victoria) gave informed, written consent for the collection of placenta (R11/34, approved 2011, research active) or an omental fat biopsy (R14/11, approved 2014, research active). Experiments were performed following institutional guidelines and regulations.

Control healthy term (delivery 37–40 weeks’ gestation) placenta, umbilical cords, and omental fat biopsies were processed within 30 min of delivery from normotensive pregnancies where a foetus of normal customized birth weight centile (>10th centile relative to gestation) was delivered [28].

### 4.2. Primary Cytotrophoblast Isolation and Culture

Human primary cytotrophoblasts were isolated from normal term placentas collected at elective Caesarean section as previously described [8,29]. In brief, placental tissue was taken from four distal sites. Tissue was mechanically minced and underwent enzymatic digestion. Cells were then separated by size using a Percoll gradient and purified via negative selection for CD9. Purified cytotrophoblast were then plated in Gibco^TM^ Dulbecco’s Modified Eagle Medium (DMEM; Thermo Fisher Scientific, Scoresby, VIC, Australia) supplemented with 10% fetal calf serum (FCS) and 1% antibiotic–antimycotic (AA; Life Technologies, Carlsbad, CA, USA) on fibronectin (10 ug/mL; BD Bioscience, San Jose, CA, USA)-coated culture plates. Cells were incubated under 8% O_2_, 5% CO_2_ at 37 °C overnight to equilibrate.

### 4.3. Human Umbilical Vein Endothelial Cell (HUVEC) Isolation and Culture

Primary HUVECs were isolated as previously described [8,30]. In brief, the umbilical cord vein from normal term placentas was cannulated and infused with phosphate buffered saline (PBS; 137 mM NaCl, 10 mM Na_2_HPO_4_, 1.8 mM KH_2_PO_4_, 2.7 mM KCl, pH 7.4) to wash out foetal blood. Approximately 10 mL (1 mg/mL) of pre-warmed collagenase (Worthington, Lakewood, NJ, USA) was infused into the cord, followed by incubation at 37 °C for 8 min. HUVECs were dissociated with an infusion of PBS and collected in newborn calf serum (Sigma Aldrich, St. Louis, MO, USA). The dissociated HUVECs were recovered by pelleting and resuspension, followed by culture in M199 media (Life Technologies, CA, USA) containing 20% newborn calf serum, 1% endothelial cell growth factor, 1% heparin (Sigma), and 1% AA, and used between passages 1–3. Cells were then seeded for treatment with M199 media containing 10% fetal calf serum, 1% endothelial cell growth factor, 1% heparin, and 1% AA and incubated under 8% O_2_, 5% CO_2_ at 37 °C overnight.

### 4.4. In Vitro Drug Treatment

Isolated primary human cytotrophoblasts and primary HUVECs were treated with 100 µM esomeprazole magnesium hydrate (MH) (E7906; Sigma) or esomeprazole magnesium trihydrate (MTH) (217087-09-7; ChemScene, Monmouth Junction, NJ, USA) in fresh media. The dose of esomeprazole used was matched to the dose-response analysis performed in our previous preclinical study [8]. Cells were incubated under placental physiological oxygen concentrations, 8% O_2_, 5% CO_2_ at 37 °C for 24 h. Cell lysates were collected for RNA extraction and media collected to assess protein secretion.

### 4.5. ELISA (Enzyme Linked Immunosorbent Assay)

Concentrations of sFLT-1 were measured in cytotrophoblast and HUVEC conditioned media using the DuoSet Human VEGF R1/FLT-1 kit (R&D systems by Bioscience, Waterloo, Australia), following manufacturer’s instructions.

### 4.6. Quantitative Real Time Polymerase Chain Reaction (qPCR)

RNA was extracted from primary cytotrophoblasts and HUVECs using the Qiagen RNeasy Mini Kit according to the manufacturer’s instructions. RNA concentration was quantified using a Nanodrop 2000 spectrophotometer (ThermoFisher Scientific, Waltham, MA, USA). RNA was converted to cDNA using the Applied Biosystems High-Capacity cDNA Reverse Transcription Kit (Thermofisher), as per manufacturer’s instructions on the iCycler iQ5 (Bio-Rad, Hercules, CA, USA).

Quantitative Taqman PCR was performed to quantify mRNA expression of *ET-1* (Hs00174961_m1; Life Technologies) and *VCAM1* (Hs01003372_m1; Life Technologies)*,* as well as reference gene, *YHWAZ* (Hs01122454_m1; Life Technologies). Taqman qPCR was performed on the CFX384 (Bio-Rad) with the following run conditions: 50 °C for 2 min, 95 °C for 10 min, 95 °C for 15 s, 60 °C for 1 min (40 cycles).

The sFLT-1 splice variants *sFLT1-i13* and *sFLT1-e15a* were measured in a SYBR PCR with SYBR Green Master Mix (Applied Biosystems) using primers specific for each variant. The primers for *sFLT1-i13* were 5′-ACAATCAGAGGTGAGCACTGCAA-3′ (forward) 5′-TCCGAGCCTGAAAGTTAGCAA-3′ (reverse), for *sFLT1-e15a*, 5-CTCCTGCGAAACCTCAGTG-3′ (forward) 5′-GACGATGGTGACGTTGATGT-3′ (reverse) and for *YWHAZ* (reference gene), 5′-GAGTCATACAAAGACAGCACGCTA-3′ (forward) 5′-TTCGTCTCCTTGGGTATCCGATGT-3′ (reverse). The SYBR PCR was run on the CFX384 (Bio-Rad), with 40 cycles of 95 °C for 21 s, then 60 °C for 20 min. All data were normalized to the reference gene as an internal control and calibrated against the average Ct of the control samples. All cDNA samples were run in duplicate.

### 4.7. Endothelial Dysfunction Rescue Studies

Using a model established previously [8], primary HUVECs were pre-treated for 2 h with 10 ng/mL tumour necrosis factor-α (TNF-α; Life Technologies), a pro-inflammatory cytokine, to induce endothelial dysfunction. Following pre-treatment, cells were then incubated (8% O_2_, 5% CO_2_ at 37 °C) for a further 24 h with 100 µM esomeprazole MH or esomeprazole MTH in the presence of TNF-α. Cell lysates were collected for RNA extraction.

### 4.8. MTS Cell Viability Assay

Cell viability was assessed following treatment using the MTS assay, CellTiter 96-AQueous One Solution (Promega, Madison WI, USA), according to manufacturer instructions. Optical density was measured using a Bio-Rad X-Mark Microplate Spectrophotometer (Hercules, CA, USA) and BioRad Microplate Manager 6 software. The assays demonstrated that treatment with both esomeprazole isomers did not affect the cell viability of HUVECs or primary cytotrophoblast cells (data not shown).

### 4.9. Reactive Oxygen Species (ROS) Assay

Isolated primary cytotrophoblasts seeded in black-walled 96-well plates were treated with 100 µM esomeprazole MH or esomeprazole MTH for 24 h (8% O_2_, 5% CO_2_ at 37 °C). Cells were then treated with ROS inducer, 1 mM sodium azide for 6 h (8% O_2_, 5% CO_2_ at 37 °C). Media was removed, then 10 µM 2′,7′–dichlorofluorescin diacetate added to stain cells for 1 h. Cells were then washed with PBS, and fluorescence measured with excitation/emission at 485 nm/520 nm on the Bio-Rad xMark Microplate Spectrophotometer.

### 4.10. Wire Myography

Omental fat biopsies were collected into ice-cold PBS, then transferred to ice cold Krebs solution (NaCl 120 mM, KCl 5 mM, MgSO_4_ 1.2 mM, KH_2_PO_4_ 1.2 mM, NaHCO_3_ 25 mM, D-glucose 11.1 mM, CaCl_2_ 2.5 mM) and kept on ice or at 4 °C until dissection. Arteries of approximately 2 mm length were isolated from surrounding fat and connective tissue. Dissected arteries were mounted on wire myograph chambers (620 M Wire Myograph, Danish Myo Technology, DMT, Hinnerup, Denmark) with 40 µm diameter tungsten wire (DMT). The chamber baths were filled with Krebs solution and bubbled continuously with carbogen (95% O_2_, 5% CO_2_). PowerLab hardware accompanied by Labchart software (ADInstruments, NSW, Australia) were used to collect and convert force measurements. The chambers with mounted vessels were incubated at 37 °C for 30 min. The vessels were normalised to 100 mmHg (13.3 kPa) pressure using the DMT normalisation module on LabChart with IC1/IC100 = 1, then allowed to equilibrate for 20 min.

Following equilibration, vessel viability was assessed. The vessels were first briefly constricted with high potassium physiological salt solution (50 mM KPSS: NaCl 75 mM, KCl 50 mM, MgSO_4_ 1.2 mM, KH_2_PO_4_ 1.0 mM, NaHCO_3_ 25 mM, D-glucose 11.1 mM, CaCl_2_ 2.5 mM) to assess smooth muscle viability. Endothelial function was confirmed by pre-constricting the vessel with the thromboxane agonist 9,11-dideoxy-9α,11α-methanoepoxy prostaglandin F2α (U46619; Sapphire Bioscience, Redfern, NSW, Australia) to 50–70% of maximal response to KPSS, followed by the addition of endothelium-dependent vasodilator bradykinin (Sapphire Bioscience). At least 80% relaxation was required to consider the endothelium intact. After 20 min of rest, the vessels were constricted with U46619 (to 50–70% of maximal response to KPSS), then treated with increasing doses (0.1–100 µM) of esomeprazole magnesium hydrate (MH) or esomeprazole magnesium trihydrate (MTH) or control (vehicle).

### 4.11. Statistical Analysis

Data were statistically tested for normal distribution and statistically tested as appropriate; compared to control with Mann–Whitney tests, and either Kruskal–Wallis or one-way ANOVA with Dunn’s correction. Myograph dose-response data were analysed using mixed-effects analysis with Dunnett’s correction, with maximal response and area under the curve analysed via Friedman’s test with Dunn’s correction. All data are expressed as mean ± SEM. *p*-Values < 0.05 were considered significant. Statistical analysis was performed using GraphPad Prism 8 software (GraphPad Software, Inc.; San Diego, CA, USA).

## Figures and Tables

**Figure 1 ijms-23-09533-f001:**
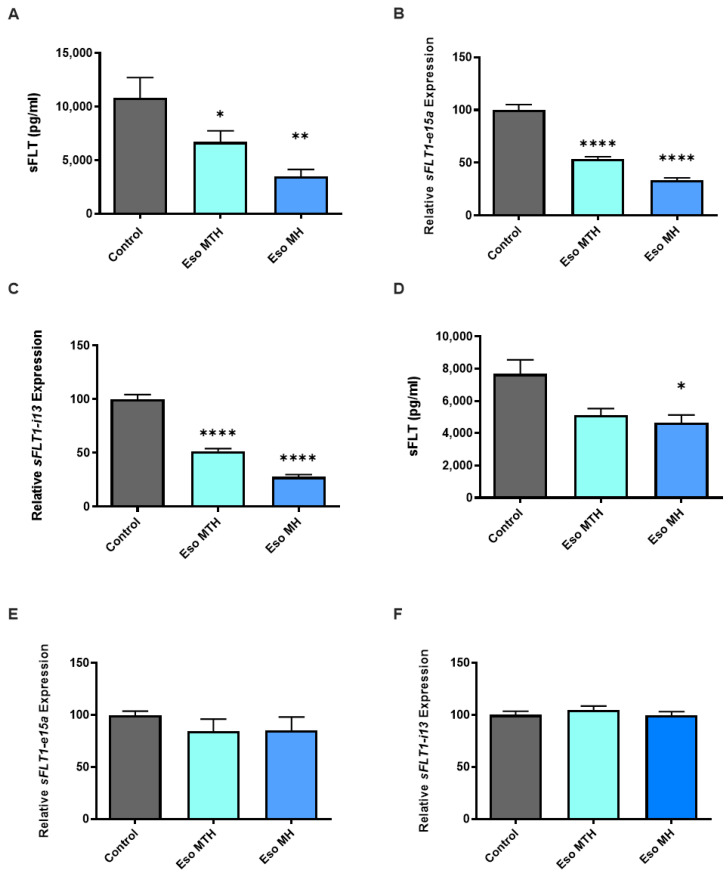
**Esomeprazole reduced expression and secretion of the antiangiogenic factor, soluble fms-like tyrosine kinase-1 (sFLT-1) in cultured human placental cells.** Esomeprazole magnesium hydrate (Eso MH) and esomeprazole magnesium trihydrate (Eso MTH) reduced secretion of sFLT-1 by primary cytotrophoblast cells (**A**). Both Eso MTH and Eso MH reduced primary cytotrophoblast mRNA expression of the sFLT-1 variants, *sFLT1-e15a* (**B**), and *sFLT1-i13* (**C**). Secretion of sFLT-1 from HUVECs was reduced by treatment with esomeprazole MH only (**D**). There was no alteration in the relative mRNA expression of *sFLT1-e15a* (**E**) and *sFLT1-i13* (**F**) in HUVECs, following treatment with either Eso MH or Eso MTH. Eso MH and Eso MTH were added at 100 μmol/L concentrations. Data expressed as mean ± SEM (*n* = 3). The statistical significance is indicated as follows: * *p* < 0.05, ** *p* < 0.01, **** *p* < 0.0001.

**Figure 2 ijms-23-09533-f002:**
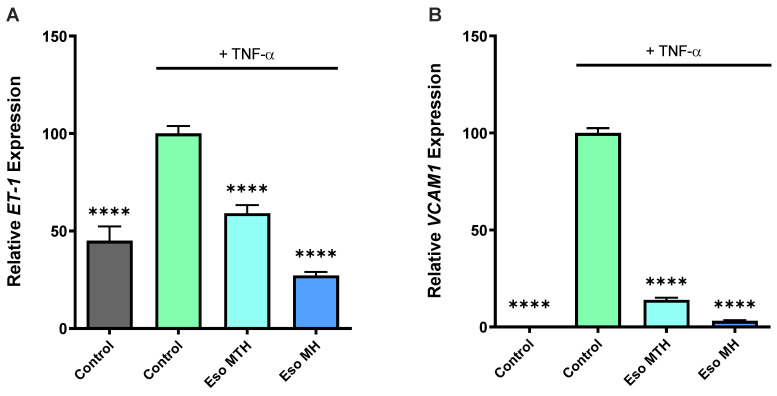
**Esomeprazole reduced expression of markers of endothelial dysfunction in response to tumour necrosis factor (TNF)-α, compared with vehicle control in primary HUVECs.** Expression of two markers of endothelial dysfunction, endothelin-1 (*ET-1*) (**A**), and vascular cell adhesion molecule-1 (*VCAM1*) (**B**) was increased in HUVECs treated with TNF-α (10 ng/mL) after 24 h (control + TNF-α, green); HUVECs not treated with TNF-α (control, grey) have significantly lower expression of markers assessed. Both esomeprazole magnesium trihydrate (Eso MTH) and esomeprazole magnesium hydrate (Eso MH) at 100 μmol/L abolished the increase in both *ET-1* (**A**), and *VCAM1* (**B**) mRNA expression in response to TNF-α. Data expressed as mean ± SEM. **** *p* < 0.0001 (*n* = 3).

**Figure 3 ijms-23-09533-f003:**
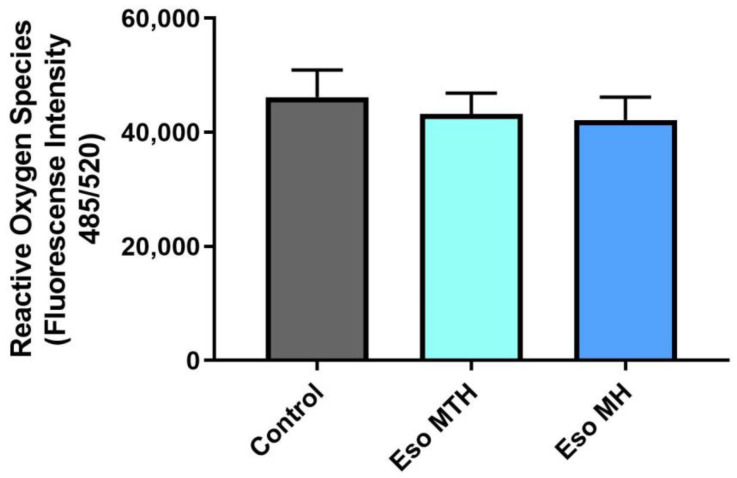
**Reactive oxygen species production induced by sodium azide (Na Azide) was not altered with treatment of either esomeprazole magnesium hydration isomer.** Reactive oxygen species production induced by Na Azide (control) was not significantly altered by treatment with esomeprazole magnesium trihydrate (Eso MTH) or esomeprazole magnesium hydrate (Eso MH). Eso MH and Eso MTH were administered at 100 μmol/L. Data expressed as mean ± SEM (*n* = 4).

**Figure 4 ijms-23-09533-f004:**
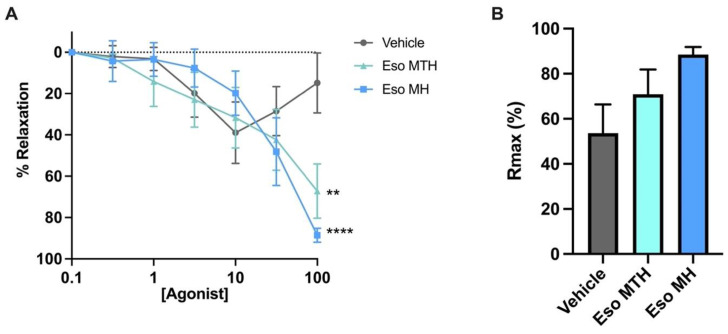
**Esomeprazole dilates human omental arteries obtained from normotensive pregnancies.** Wire myograph assessment of arterial reactivity following constriction with the vasoconstrictor U44619 (thromboxane agonist), and treatment with 0.1–100 μmol/L esomeprazole magnesium trihydrate (Eso MTH) and esomeprazole magnesium hydrate (Eso MH), compared with vehicle (**A**,**B**). Data presented as percentage of relaxation (% relaxation) following normalisation to the maximum relaxation to bradykinin, demonstrating both Eso MTH and Eso MH significantly relaxed vessels at the highest concentration (100 μmol/L) (**A**). The maximum relaxation induced by Eso MTH or Eso MH show no significant difference to vehicle treated vessels (**B**). Data expressed as mean ± SEM; significance is indicated by the following ** *p* < 0.01, **** *p* < 0.0001 (*n* = 7). The *x*-axis is in log10 scale.

## Data Availability

Data available upon reasonable request from corresponding author.

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
