# Peer review of "Assessment of the Proton Pump Inhibitor, Esomeprazole Magnesium Hydrate and Trihydrate, on Pathophysiological Markers of Preeclampsia in Preclinical Human Models of Disease"

_ijms, 2022, doi:10.3390/ijms23179533_

Round 1

Reviewer 1 Report

The title must be changed as it is not allowed to prepare a “human model” of a disease. One can  perform in-vivo study in a group of patients or healthy subjects or a study using  animal models , or a study using human samples  - blood or other tissues,  or in-vitro study using cell-lines.  Thee is a study in which human tissues were used to perform in-vitro experiments. It must be underlined in the title, that there are data from a preclinical study.  

Line 93, line 134  – “As we had showed previously´ “As observed previously - it suggest that not all data presented on Fig. 1/2 are from the same experiment, and part of the presented data were published before. It must be clarify. Maybe the data are consistent with data of your previous studies, but, if so,  a reference should be added.

Fig.2 – there are to Controls and it is not explained in the figure legend. Therefore, to have two Controls will make readers confused.  

Reviewer 2 Report

The work appears to be well done, but the presentation in many places is careless. The manuscript would benefit from addressing the following points:

1.      Authors should clearly indicate the concentrations of MH and MTH used for the experiments in the Abstract and Materials and Method, as well as to explain the rationale of using these concentrations (probably, cite the previous study if the dose-response analysis was performed earlier).

2.      Legend to Figure 1: the legend “Secretion of sFLT-1 from HUVECs was reduced by treatment with both esomeprazole compounds (D)” contradicts the results presented on the graph and in the text (lines 105-106).

3.      Subsection 2.3 and Legend to Figure 3: please, carefully check the text since the following sentences are unclear and appear to contradict the results presented on the graphs:

Lines 157-158: “Cytotrophoblasts that were pre-incubated with esomeprazole MTH demonstrated significantly increased ROS (Figure 3, p=0.048)” (probably, “in the presence of Na Azide” should be added).

Lines 167-169: “elevated production of ROS by Na Azide was blocked with preincubation with esomeprazole magnesium hydrate (Eso MH), however remained unchanged in response to esomeprazole magnesium trihydrate (Eso MTH)”. This conclusion is questionable since the statistical significance was established in comparison to control, however, in this case the significance between Na azide alone, Na azide + ESO MTH and Na azide + ESO MH should be calculated. Looking at the graphs, I don’t see considerable difference between these values.

4.      Lines 192 and 197: “compared to vehicle (A-C)” – there is no panel C on the graph.

5.      Figure 4A: please, replace the scale of X axis from (-1-2) to (0,1-100).

6.      Subsection 4.8. Where are the results of MTS cell viability assay?

7.      If the conclusion “Esomeprazole MH in general appears to be more efficacious in our in vitro assays, compared to esomeprazole MTH” is made, the differences between values obtained for MH- and MTH-treated cells should be evaluated for statistical significance.

Round 2

Reviewer 1 Report

The manuscript was found significantly improved according the previous comments.

No additional comments. The manuscript was found suitable for publication